# Thermal Decomposition Study on Li_2_O_2_ for Li_2_NiO_2_ Synthesis as a Sacrificing Positive Additive of Lithium-Ion Batteries

**DOI:** 10.3390/molecules24244624

**Published:** 2019-12-17

**Authors:** Jaekwang Kim, Hyunchul Kang, Keebum Hwang, Songhun Yoon

**Affiliations:** Department of Nanomaterials Science and Engineering, School of Integrative Engineering, Chung-Ang University, 84, Heukseok-ro, Dongjak-Gu, Seoul 06974, Korea; kjk9025@hanmail.net (J.K.); guscjf763@gmail.com (H.K.); hwang1443@naver.com (K.H.)

**Keywords:** Li_2_O_2_, thermal decomposition, Li_2_O, Li_2_NiO_2_, Li source, Li-ion battery

## Abstract

Herein, thermal decomposition experiments of lithium peroxide (Li_2_O_2_) were performed to prepare a precursor (Li_2_O) for sacrificing cathode material, Li_2_NiO_2_. The Li_2_O_2_ was prepared by a hydrometallurgical reaction between LiOH·H_2_O and H_2_O_2_. The overall reaction during annealing was found to involve the following three steps: (1) dehydration of LiOH·H_2_O, (2) decomposition of Li_2_O_2_, and (3) pyrolysis of the remaining anhydrous LiOH. This stepwise reaction was elucidated by thermal gravimetric and quantitative X-ray diffraction analyses. Furthermore, over-lithiated lithium nickel oxide (Li_2_NiO_2_) using our lithium precursor was synthesized, which exhibited a larger yield of 90.9% and higher irreversible capacity of 261 to 265 mAh g^−1^ than the sample prepared by commercially purchased Li_2_O (45.6% and 177 to 185 mAh g^−1^, respectively) due to optimal powder preparation conditions.

## 1. Introduction

Within Li-ion (Li^+^) batteries (LIBs), organic electrolytes are easily decomposed due to the highly negative potential of the anode materials during first few cycles. This decomposition can be favorable in one sense, since the solid electrolyte interphase (SEI) passivates the surface of the electrode and prevents further decomposition of the electrolyte. The stabilized SEI film resulted in reasonable coulombic efficiency and cycle performance [1,2,3,4]. However, the electrolyte decomposition is not beneficial with respect to the specific energy of LIBs. Since decomposition consumes electric charges and Li^+^ during the first few charging cycles, an overload of the positive electrode material is inevitable to compensate the Li^+^ and electric charges for the SEI formation [5,6]. The carbonaceous materials, such as soft carbons, hard carbons, graphite, etc. have been widely used as anode materials for LIBs for recent device applications such as electric vehicle (EV) and hybrid EV [7,8,9,10,11,12,13,14,15]. At present, moreover, high-capacity materials are on use (ex. SiO) in commercial LIBs to increase energy density. Unlike to a graphite (>90%), the SiO exhibits a quite low coulombic efficiency about 45% in the initial few cycles because of a charge consumption originated from the irreversible formation of the SEI. However, this high-capacity material have been applied to meet the demands of higher energy density cells. Then, the compensation of charge (or Li^+^) consumption should become a critical issue in the future market of LIBs [3,16,17]. For this reason, the role of a sacrificing positive electrode additive has been widely discussed. Researchers who studied the sacrificing materials reported that the materials should be of a light weight for better energy density, easily decomposed near the charging potential of cathode material, and over-lithiated for maximum irreversible capacity to match the consumed charges (and Li^+^).

The over-lithiated lithium nickel oxide (Li_2_NiO_2_) has been considered as a promising additive since it is both over-lithiated to produce the large irreversible capacity during initial cycles and of light weight [18]. Li_2_NiO_2_ is widely synthesized by nickel oxide (NiO) and lithium oxide (Li_2_O) since they directly react in solid-state in stoichiometric ratio 1:1. However, as the main precursor of Li_2_NiO_2_, Li_2_O is still expensive to prepare as it is obtained by direct oxidizing of Li metal. Therefore, the efficient and economic preparation of the Li_2_O is highly required. The thermal decomposition of lithium peroxide (Li_2_O_2_) is can be less expensive way to synthesize Li_2_O, because it only exhales O_2_ during its reaction and its precursor (LiOH·H_2_O) is inexpensive. To the best of our knowledge, the systematic study of this thermal decomposition of Li_2_O_2_ has not yet been reported. To provide a better insight into this beneficial synthesis method, the authors have investigated the thermal decomposition of Li_2_O_2_ by applying various decomposition temperatures and reaction times. Quantitative analysis is performed on the reaction products to elucidate the composition ratios of several lithium compounds obtained from the reaction. Moreover, two types of Li_2_NiO_2_ (L2N) are obtained by using as prepared or commercially purchased Li_2_O, and their electrochemical performances are compared.

## 2. Results and Discussion

The morphologies of the Li_2_O_2_ and its thermally decomposed products at 450 °C and 600 °C are characterized by the FE-SEM images presented in Figure 1. The 450 °C reaction products display well-defined primary particles compared to the Li_2_O_2_. However, as-prepared particles tended to aggregate, suggesting that their surfaces were slightly sintered to produce a massive secondary particle. In support of this observation, the secondary particles reacted at 600 °C displayed a highly sintered phase and it is difficult to find any primary particle boundaries. Since the Li_2_O_2_ was synthesized by the hydrometallurgical reaction between LiOH·H_2_O and H_2_O_2_ to generate Li_2_O_2_·H_2_O_2_·3H_2_O, contamination with LiOH·H_2_O or anhydrous LiOH during vacuum, drying is inevitable [19]. Furthermore, it is widely known that LiOH·H_2_O and anhydrous LiOH are easily sintered when transformed to the anhydrous phase and Li_2_O, respectively, via the molten phase [20]. Therefore, it is obvious that an unintended Li impurity exists in the prepared Li_2_O_2_ and that its presence should be attributed to the monolithic phase of a reacted product during the thermal decomposition.

For further investigation, XRD and TGA analyses were conducted on the prepared materials. The XRD spectra of the materials prepared at 350, 450, 600 °C are presented in Figure 2a. The materials were exposed to individual reaction temperature for 60 min. The red, navy and yellow columns represent the XRD peak indices of Li_2_O_2_ (PDF-01-073-1640), LiOH (PDF-01-076-0911) and LiOH·H_2_O (PDF-01-076-1073), respectively. The vertical dotted lines indicate the locations of Li_2_O (lithia) peaks with an Fm3m cubic structure (PDF-01-076-9262). Since dehydration of LiOH·H_2_O occurs between 90 and 200 °C, there is no trace of an LiOH·H_2_O crystalline signal in any of the spectra; instead, anhydrous LiOH peaks are observed on each XRD profile [21,22]. As the reaction temperature increased, the Li_2_O_2_ remaining in the 350 °C sample disappeared, the majority of it being transformed into Li_2_O.

The TGA and differential thermal gravimetric (DTG) results are also plotted in Figure 2b to elucidate the change in the Li_2_O_2_ sample during thermal decomposition. The analysis was performed between room temperature and 900 °C at a heating rate of 5 °C min^−1^. The full reaction consists of the following three steps: (1) dehydration of LiOH·H_2_O, (2) decomposition of Li_2_O_2_, and (3) pyrolysis of anhydrous LiOH, as indicated by arrows in the plot. At the beginning of the DTG profile, a minute amount of signal was detected between 90 and 200 °C, the temperature at which the dehydration of LiOH·H_2_O occurs [21,22]. Note that this is consistent with the FE-SEM results for the sintered material and absence of LiOH·H_2_O peaks in the aforementioned XRD. The Li_2_O_2_ decomposition started at 340 °C and ended at 420 °C, which is in agreement with the results of the other group who studied the decomposition of Li_2_O_2_ [23,24]. A broad and small signal appearing between 420 and 550 °C was probably related to the pyrolysis of the anhydrous LiOH reported by Kiat et al. [25]. However, since their results were obtained using a vacuum annealing atmosphere (which is different from our N_2_ atmosphere), this would need to be verified.

In order to verify the full decomposition of Li_2_O_2_, several thermal annealing experiments were conducted. The gravimetric composition of an obtained Li compound was quantitively analyzed via XRD Rietveld refinement, as listed in Table 1. While a large amount of unreacted Li_2_O_2_ remained in the 350 °C samples, the content of Li_2_O nevertheless increased with increasing thermal exposure time. On the other hand, the Li_2_O_2_ reacted thoroughly after 60 min at 400 °C to form the other Li compounds. Even if the quantity of Li hydroxides deviates, however, it is noteworthy that they were not found after the reaction at 500 °C, as expected from the TGA and DTG results. Moreover, the reaction trend of the Li_2_O shows consistency irrespective of the hydroxides. Thus, the gravimetric information for Li_2_O in Table 1 was converted to mole fraction of reacted product and listed in Table 2 for detailed investigation.

The mole (phase) fraction of Li_2_O within a decomposed product was plotted against exposure time at each reaction temperature, as shown in Figure 3a. The linear profiles depicted over the dotted results represent the best-fitted results of thermal conditions using the least-square method. While the line surges adversely to a high phase fraction at 350 °C, the others rise somewhat slowly. Since their slopes equal the reaction rate of Li_2_O in Equation (1), the reaction rate was plotted against the reaction temperature for better insight:(1)$$Li_{2}O_{2}\left(s\right)\;\to Li_{2}O\left(s\right)\;+\frac{1}{2}O_{2}\left(g\right)$$

The profile in Figure 3b shows excessive reaction between 350 and 400 °C, whereas the other regions exhibit small and broad reaction profiles. Moreover, these phenomena and fitted slopes agree well with the DTG profiles in Figure 2b. More interestingly, the small and broad reaction profile in Figure 3b can be attributed to the pyrolysis of Li hydroxides, since a slight trace of hydroxide compounds was found after the samples were annealed at 600 °C. Consequently, it can be said that the broad and small peak observed in Figure 2b originated from the pyrolysis of anhydrous LiOH under an N_2_ atmosphere. To summarize the full decomposition reaction, the transformation ratios of Li_2_O presented in Table 2 were plotted on a three-dimensional graph (Figure 3c) with each color division representing a mole fraction of 20%.

To estimate the performance of the prepared Li_2_O (decomposed at 450 °C for 10 h under N_2_ flow) as a precursor of the L2N sacrificing additive material of a LIB cathode, two types of L2N compound were synthesized using the prepared material (P-L2N) and commercially purchased material (C-L2N). Spherical shaped NiO particles were used for the synthesis. Appendix A shows the SEM images of the two L2N samples. It appears that more sintered particles are found in the P-L2N compound than in the C-L2N. Moreover, spherical NiO particles are more abundant in the SEM image of the C-L2N presented in Appendix A. XRD was performed on the synthesized L2N and the diffraction patterns are presented in Appendix A. While both materials exhibited the characteristic peaks of L2N, there may be some difference in signal intensity. Nevertheless, residual NiO and Li_2_O signals are observed for both materials. Therefore, Rietveld refinement was conducted on both XRD patterns to indicate that 90.9% and 45.6% of L2N exist in the P-L2N and C-L2N, respectively.

Figure 4 presents the galvanostatic charge/discharge profiles of the prepared L2N, as recorded between 3.0 and 4.25 V with a 0.1 C current injection. For the first cycle of charge/discharge, three identically conditioned cells were tested for repeatability. As a promising sacrificing positive additive providing surplus Li-ion to a counter (negative) electrode, an irreversible capacity of the first cycle is generally considered as a key point index for the L2N material [3]. The P-L2N and C-L2N showed 261 to 265 and 177 to 185 m Ah g^−1^ of irreversible capacity at the first cycle. Therefore, the yield of the L2N synthesis can be regarded as significant for its performance. In this regard, the Li_2_O material produced by thermal decomposition of Li_2_O_2_ can be beneficial for the L2N additive synthesis.

## 3. Materials and Methods

### 3.1. Preparation of Lithium Peroxide (Li_2_O_2_)

Li_2_O_2_ powder was prepared by a simple precipitation method. For the precipitation, lithium hydroxide monohydrate (1.25 g, LiOH·H_2_O, Junsei Chemical Co. Ltd., Tokyo, Japan) was poured into 100 mL of vigorously stirred hydrogen peroxide (30% H_2_O_2_ (aq), Samchun Chemicals, Seoul, Korea). The stirring was continued for an hour until a yellowish precipitate was obtained. The precipitate was filtered using a cellulose filter paper and dried in a vacuum oven under ambient temperature conditions until parched.

### 3.2. Thermal Decomposition of Lithium Peroxide (Li_2_O_2_) to Lithium Oxide (Li_2_O)

The Li_2_O_2_ powder (10 g) was placed in an alumina (Al_2_O_3_) crucible and heated in a tube furnace with a continuous flow of N_2_ (99.999%) gas into the tube. The ramping rate was 5 °C/min and the reaction temperature was 350–600 °C. After reaching the reaction temperature, the thermal exposure time was varied from 30 to 120 min to find the economically optimal reaction time while obtaining highly pure Li_2_O.

### 3.3. Preparation of the L2N

The Li_2_NiO_2_ (L2N) was synthesized by a solid-state reaction described elsewhere [3,18]. Two type of L2N were synthesized by using different Li_2_O samples (as prepared and commercially purchased sample). For this synthesis, we decomposed the Li_2_O_2_ at 450 °C for 2 h. Stoichiometrically mixed NiO and Li_2_O were placed in an alumina crucible and heated at 650 °C for 12 h under an N_2_ atmosphere. The obtained L2N powder was handled under an argon atmosphere in a glove box because it would be unstable in air.

### 3.4. Material Characterization

The Morphology of the material was characterized by field emission scanning electron microscopy (FE-SEM 500, ZEISS Sigma, Oberkochen, Germany). In addition, crystallographic analysis was conducted by X-ray diffraction (New D8-advance, Bruker-AXS, Karlsruhe, Germany) from 0 to 130 degrees and the quantitative analysis was performed by HighScore (Malvern Panalytical, Great Malvern, UK). Moreover, thermalgravimetric analysis (TGA) was conducted on the prepared Li_2_O_2_.

### 3.5. Electrochemical Analysis

For the electrochemical performance estimation, a 2032 type coin cell was used as a half-cell. An active material (L2N), binder and conductive agent (carbon black) were homogeneously mixed in N-methyl-2-pyrrolidone (NMP) solvent to form a slurry, which was then coated onto an aluminum foil current collector, pressed and dried overnight in a vacuum oven at 120 °C. The 2032 type coin cells were assembled with the Li_2_NiO_2_ working electrode, Li foil counter electrode and 1.0 M LiPF_6_ in ethylene carbonate (EC)/ethyl methyl carbonate (EMC) (1:2 *v*/*v*%) electrolyte. The cells were assembled under an argon atmosphere (H_2_O: ~0.1 ppm) in a glove box. After assembly, the cells were rested for 10 h.

To measure the electrochemical performance of the assembled cells, galvanostatic charge-discharge was conducted on three cells for each material with constant-current/constant-voltage (CCCV) mode between a potential window of 3.0–4.25 V (vs. Li/Li^+^) by applying a 0.1 C current rate.

## 4. Conclusions

In summary, various thermal decomposition experiments of Li_2_O_2_ were conducted to find the optimal reaction conditions to produce Li_2_O. The full reaction during annealing was found to involve the following steps: (1) dehydration of LiOH·H_2_O, (2) decomposition of Li_2_O_2_ and (3) pyrolysis of the remaining anhydrous LiOH. This stepwise reaction was elucidated by the TGA, and Rietveld refinement analysis using XRD. Using the prepared precursor and commercial one, two types of Li_2_NiO_2_ were synthesized and the Li_2_NiO_2_ synthesized from the prepared Li_2_O showed a larger yield of 90.9% and higher irreversible capacity of 261 to 265 mAhg^−1^ than the sample synthesized by commercially purchased Li_2_O (45.6% and 177 to 185 mAh g^−1^, respectively). Consequently, it was confirmed that the Li_2_O decomposed from Li_2_O_2_ was highly suitable precursor in the sacrificing cathode material.

## Figures and Tables

**Figure 1 molecules-24-04624-f001:**
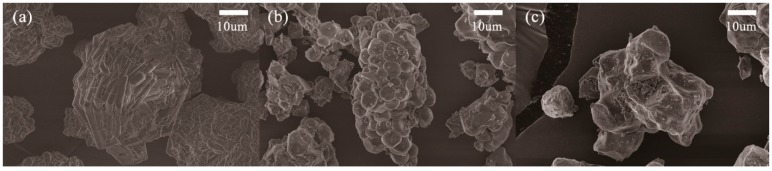
FE-SEM images of (**a**) Li_2_O_2_ and its decomposed products at (**b**) 450 °C and (**c**) 600 °C.

**Figure 2 molecules-24-04624-f002:**
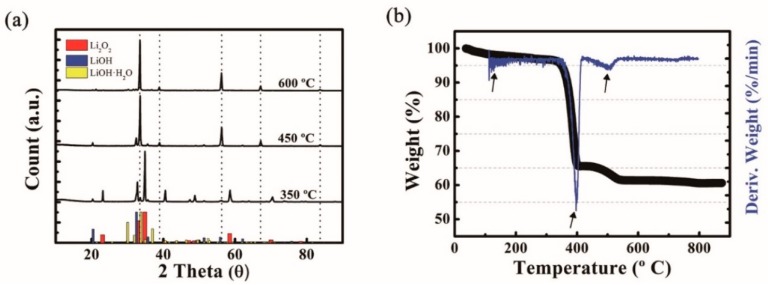
(**a**) XRD spectra of the materials decomposed at 350, 450, 600 °C and (**b**) TGA (black) and differential thermal gravimetric (DTG, navy color).

**Figure 3 molecules-24-04624-f003:**
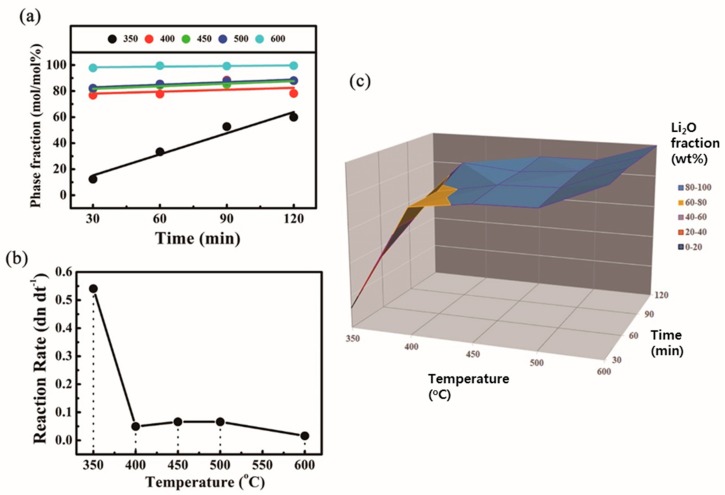
(**a**) Mole fraction of Li_2_O within a decomposed product according to reaction time. (**b**) Reaction rate of each exposure temperature derived from the slope of mole fraction of Li_2_O. (**c**) 3D graphical summary of Li_2_O transformation ratio (wt%) obtained from Table 1.

**Figure 4 molecules-24-04624-f004:**
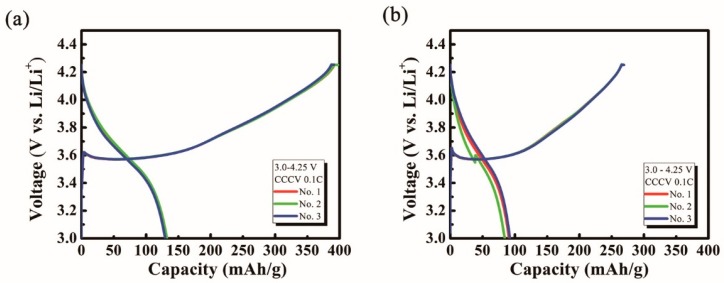
Galvanostatic charge/discharge profiles obtained from three identically conditioned cells of (**a**) P-L2N and (**b**) C-L2N.

**Table 1 molecules-24-04624-t001:** Total gravimetric ratio of the Li compound after thermal reaction.

Temp (°C)	Time (min)	Li_2_O_2_	Li_2_O	LiOH	LiOH·H_2_O	Li_2_CO_3_	Sum (wt%)
**350**	30	85.7	2.1	11.7	0.5	0	100
	60	84.7	5.8	9.2	0.2	0	99.9
	90	75.9	14.3	10.1	0.3	0	100
	120	70.7	18.9	9.9	0.4	0	99.9
**400**	30	25.8	59.5	14.0	0.5	0.2	100
	60	0.1	80.1	17.4	2.3	0.2	100.1
	90	0.2	89.2	7.8	2.6	0.3	100.1
	120	0	81	16.9	2.1	0	100
**450**	30	0	82.4	11.3	6.4	0	100.1
	60	0	83.5	12.2	0.3	0	100
	90	0	87.5	12.2	0.3	0	100
	120	0	89.5	9	1.5	0	100
**500**	30	0	82.8	10.5	6.7	0	100
	60	0	85.9	8.7	5.4	0	100
	90	0	87.7	6.3	6	0	100
	120	0	87.7	5.9	6.4	0	100
**600**	30	0	97.9	1.6	0	0.5	100
	60	0	99.4	0.2	0.4	0	100
	90	0	99.1	0.2	0.8	0	100.1
	120	0	99.5	0.2	0.3	0	100

**Table 2 molecules-24-04624-t002:** Mole fraction of Li_2_O among the product of thermal reaction (mol%).

	350 °C	400 °C	450 °C	500 °C	600 °C
**30 min**	12.3	76.8	81.5	82.2	97.8
**60 min**	33.3	77.7	84.4	85.4	99.5
**90 min**	52.7	88.4	85.0	87.8	99.2
**120 min**	59.9	78.2	87.9	88.0	99.5

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
