# Peer review of "Thermal Decomposition Study on Li2O2 for Li2NiO2 Synthesis as a Sacrificing Positive Additive of Lithium-Ion Batteries"

_molecules, 2019, doi:10.3390/molecules24244624_

Round 1

Reviewer 1 Report

Thermal Decomposition of Li2O2 for Li2NiO2 2 Synthesis as a Sacrificing Positive Additive of Lithium-Ion Batteries was studied in this paper. However, the novelty of this paper is poor and need major revision.

The introduction is not fully reviewed, and the purpose is not clear of this paper.

In table 1, why some of the sum value exceed 100?

The unit in figure 3 (c) was missed.

In figure 4, why the irreversible capacity of the first cycle is too large?

Where is the conclusion?

Author Response

Reviewer #1

Comments and Suggestions for Authors

Thermal Decomposition of Li2O2 for Li2NiO2 Synthesis as a Sacrificing Positive Additive of Lithium-Ion Batteries was studied in this paper. However, the novelty of this paper is poor and need major revision.

The introduction is not fully reviewed, and the purpose is not clear of this paper.

Thank you for your considerate opinion on our submitted manuscript. We agree the reviewer’s opinion about the wrong introduction section and have edited the introduction section as the reviewer pointed out. Also, we gently emphasized the purpose of our research.  

In table 1, why some of the sum value exceed 100?

As the reviewer kindly pointed out, some of the sum values exceeds or lacks than 100 % since they are obtained by computer program for automatic Rietveld refinement (Highscore, Malvern Panalytical). But, we think that these minute program errors would be okay since they doesn’t interfere our findings.  

The unit in figure 3 (c) was missed.

Thank you about the reviewer’s kind mention about our mistake and we have added the caption letter (c) in the Figure 3.  

In figure 4, why the irreversible capacity of the first cycle is too large?

There is some reason that the reviewer pointed out, however, as mentioned in our introduction section, the Li2NiO2 material is generally used as sacrificing additive of cathode. This additive material compensates irreversible capacity of anode during first or first few cycles, since the solid electrolyte interphase (SEI) is formed on the surface of an electrode by consuming electric charges and Li-ions (Li+). Therefore, we think that the large irreversible capacity is natural phenomena on this research.

Where is the conclusion?

There are some reasons that the reviewer pointed out, the “instructions for authors” said that the conclusions are optional, moreover we merely wanted to provide the researchers (or manufacturers) with the full information about the decomposition conditions and let them choose a condition which they need. However, we too agree about the reviewer’s question, since we think that a research article needs conclusion. Therefore, we have added conclusion section as a form of summary and reason for the Li2O2 usage for Li2O preparation.

Reviewer 2 Report

The manuscript entitled "Thermal Decomposition Study on Li2O2 for Li2NiO2 Synthesis as a Sacrificing Positive Additive of Lithium-Ion Batteries" reports a work in the world of electrodes.
Before to accept the present manuscript the authors should be revise the English language, explain the innovation of this work, focus the introduction section on the recent investigations on these cathode electrodes for this kind of applications and more details about the sample preparation.

Author Response

Reviewer #2

The manuscript entitled "Thermal Decomposition Study on Li2O2 for Li2NiO2 Synthesis as a Sacrificing Positive Additive of Lithium-Ion Batteries" reports a work in the world of electrodes.

Before to accept the present manuscript the authors should be revise the English language, explain the innovation of this work, focus the introduction section on the recent investigations on these cathode electrodes for this kind of applications and more details about the sample preparation.

Thank you for your valuable opinion on our submitted manuscript. There are some reasons that the reviewer mentioned about the revision of the English language, however, our manuscript has already been edited by a professional translator, therefore, we have enveloped the certificate that proves this issue with our revised manuscript. We agree the reviewer’s opinion about the wrong introduction section and have edited the introduction section as the reviewer pointed out. Also, we gently emphasized the reason of our research to explain the innovation of this work.

We agree with the reviewer’s opinion on the details about the sample preparation and modified our manuscript, however, it may seem minute since the preparation process itself is literally simple.

Round 2

Reviewer 1 Report

The unit in figure 3 (c) is still missed.

Reviewer 2 Report

All comments have been introduced in the manuscript